# New Insight into Aspartate Metabolic Pathways in *Populus*: Linking the Root Responsive Isoenzymes with Amino Acid Biosynthesis during Incompatible Interactions of *Fusarium solani*

**DOI:** 10.3390/ijms23126368

**Published:** 2022-06-07

**Authors:** Mei Han, Xianglei Xu, Xue Li, Mingyue Xu, Mei Hu, Yuan Xiong, Junhu Feng, Hao Wu, Hui Zhu, Tao Su

**Affiliations:** 1Co-Innovation Center for Sustainable Forestry in Southern China, College of Biology and the Environment, Nanjing Forestry University, Nanjing 210037, China; sthanmei@njfu.edu.cn (M.H.); 1736164453@njfu.edu.cn (X.X.); xueliforest@njfu.edu.cn (X.L.); mingyuexu0116@163.com (M.X.); humei@njfu.edu.cn (M.H.); xiongyuan@njfu.edu.cn (Y.X.); domonor@163.com (J.F.); wh000@njfu.edu.cn (H.W.); zh8201711509@njfu.edu.cn (H.Z.); 2Key Laboratory of State Forestry Administration on Subtropical Forest Biodiversity Conservation, Nanjing Forestry University, Nanjing 210037, China; 3Key Laboratory of Plant Functional Genomics and Resources, Shanghai Chenshan Botanical Garden, Shanghai 201602, China

**Keywords:** *Populus*, nitrogen metabolism, aspartate pathway, amino acids, defense response

## Abstract

Integrating amino acid metabolic pathways into plant defense and immune systems provides the building block for stress acclimation and host-pathogen interactions. Recent progress in L-aspartate (Asp) and its deployed metabolic pathways highlighted profound roles in plant growth and defense modulation. Nevertheless, much remains unknown concerning the multiple isoenzyme families involved in Asp metabolic pathways in *Populus trichocarpa*, a model tree species. Here, we present comprehensive features of 11 critical isoenzyme families, representing biological significance in plant development and stress adaptation. The in silico prediction of the molecular and genetic patterns, including phylogenies, genomic structures, and chromosomal distribution, identify 44 putative isoenzymes in the *Populus* genome. Inspection of the tissue-specific expression demonstrated that approximately 26 isogenes were expressed, predominantly in roots. Based on the transcriptomic atlas in time-course experiments, the dynamic changes of the genes transcript were explored in *Populus* roots challenged with soil-borne pathogenic *Fusarium solani* (Fs). Quantitative expression evaluation prompted 12 isoenzyme genes (*PtGS2/6*, *PtGOGAT2*/*3*, *PtAspAT2*/*5*/*10*, *PtAS2*, *PtAspg2*, *PtAlaAT1*, *PtAK1*, and *PtAlaAT4*) to show significant induction responding to the Fs infection. Using high-performance liquid chromatography (HPLC) and non-target metabolomics assay, the concurrent perturbation on levels of Asp-related metabolites led to findings of free amino acids and derivatives (e.g., Glutamate, Asp, Asparagine, Alanine, Proline, and α-/γ-aminobutyric acid), showing marked differences. The multi-omics integration of the responsive isoenzymes and differential amino acids examined facilitates Asp as a cross-talk mediator involved in metabolite biosynthesis and defense regulation. Our research provides theoretical clues for the in-depth unveiling of the defense mechanisms underlying the synergistic effect of fine-tuned Asp pathway enzymes and the linked metabolite flux in *Populus*.

## 1. Introduction

A wide range of pathogenic microorganisms (e.g., fungi, bacteria, and viruses) represent a critical threat to the survival of forest tree species. Remarkably, fungal disease exerts a fundamental role in plant evolution, ecology, food biomass, and bioenergy [1]. Genetic variation, pathogen virulence, and favorable environments have long been regarded as affecting patterns of disease incidence and outbreaks in plant hosts. Coherently, variable environmental conditions, coordinated with nutrition factors, potentially impact the abundance of phytopathogens [2,3]. In various plant species, identifying natural nutritional molecules that can improve biotic stress tolerance has become a well-recognized and promising strategy for mitigating pathogen damage by regulating metabolic processes [4]. Thus, there is a great need for efficient use of dynamic environmental cues to unravel multidimensional mechanisms of plant–pathogen interactions and adapt these to sustainable forests and ecological systems which are resilient to global warming and climate change.

As the key organic nitrogen (N) form, L-aspartate (Asp) is an intrinsically endogenous metabolic limiter that enables cell proliferation when respiration is hampered, relevant to therapeutic targets [5]. Also, representing one of the significant intra- and inter-cellular N carriers in plants, Asp is generally transported in the vascular tissues and facilitates the bioactive compound and critical precursor [6]. Asp has been well documented as playing the core metabolite hub, interconnecting the biosynthesis of four essential Asp-family amino acids, lysine (Lys), threonine (Thr), methionine (Met), and isoleucine (Ile), aromatic amino acids, and functional intermediates, including organic acids, nicotinamide adenine dinucleotide (NAD), and purine nucleotides [7,8]. Asp oxidation and subsequent pyridine nucleotide biosynthesis exert a profound metabolic route for plant disease resistance upon pathogen invasion. Besides this, Asp participates in glycolysis, phytohormone conjugation (e.g., indole-3-acetic acid (IAA) and ethylene), and cross-talk between salicylic acid (SA) and jasmonic acid (JA), pointing out the primary features for plant growth and immune control through N recycling, translocation, and signaling [9]. The physiological impacts of Asp in plants have been reviewed previously, uncovering the conspicuous roles of Asp in regulating plant adaptation and tolerance to abiotic and biotic stress cues [8]. Asp biosynthesis and its associated diverse metabolic pathways involve the molecular modulation of multiple enzyme families, linking a broad amino acid and carbohydrate metabolism [10].

Regarding N utilization, the upstream ammonium (NH_4_^+^) transported in roots is firstly assimilated into organic glutamine (Gln) by Gln synthetase (GS, EC 6.3.1.2). The transfer of the Gln-amid group to 2-oxoglutarate (2-OG) is conducted by glutamate (Glu) synthase (GOGAT, Gln: 2-OG amidotransferase), yielding two molecules of Glu. Two distinct forms of GOGAT are present in higher plants with the subcellular target in plastids: Fd-GOGAT (EC 1.4.7.1) uses reduced ferredoxin (Fd) as the electron donor, and NADH-GOGAT (EC 1.4.1.14) uses NADH as the electron donor [11]. Relying on the GS/GOGAT cycle, Gln and Glu are consistently synthesized as N and carbon (C) sources and are further metabolized into amides and other amino acids. The Asp aminotransferase (AspAT, EC 2.6.1.1) exerts the primary role in association with amino acid biosynthesis and shuttling C skeletons from the tricarboxylic acid (TCA) cycle. The AspAT family is characterized by different cellular organisms, including prokaryotic and eukaryotic. The eukaryotic AspAT hydrolyzes the de novo Asp synthesis through a reversible transamination of Glu with oxaloacetate (OAA). The prokaryotic AspAT, so-called PATs/PPA-ATs (prephenate aminotransferase, EC 2.6.1.78), catalyze the conversation of Asp and prephenate to arogenate, a precursor used for aromatic tyrosine (Tyr) and phenylalanine (Phe) biosynthesis [12]. Recently, the resulting enhanced Asp accumulation from increased glucose consumption during cellular hypertrophy provided the hint that glucose is the essential C carrier, through the intermediate of pyruvate into the TCA cycle, directing Asp biosynthesis [13].

In plants, asparagine synthetase (AS/ASN, EC 3.5.1.1) transfers the Glu-amide group or NH_4_^+^ into Asp, the substrate for asparagine (Asn) biosynthesis. Inversely, asparaginase (Aspg, EC 6.3.1.2) catalyzes the Asn-amide group to release NH_4_^+^ and Asp. The hydrolysis of the Asn-amide group with Asn is intermediated by Asn aminotransferase (AsnAT, EC 2.6.1.45), producing NH_4_^+^ and 2-oxosuccinamate (2-OS). Subsequently, AspAT deprives the latter product of being converted into OAA and further consumed in the Glu/Asp cycle. The reaction transferring amino between serine (Ser) and glyoxylate is designated as AGT1, owing to its homology to amino Ala: glyoxylate aminotransferase 1 (AGT1, EC 2.6.1.44) in mammalians [14]. Using variable amino acceptors as substrates, AsnAT also catalyzes Asn transamination into other amino acids, including glycine (Gly), alanine (Ala), Ser, and homoserine (homo-Ser) [15]. Ala aminotransferase (AlaAT, EC 2.6.1.2) catalyzes the dual-conversion of Ala to pyruvate by consuming 2-OG, yielding Glu that is integrated for Asp biosynthesis. Another homologous subfamily encoding for AlaAT acts as Glu: glyoxylate aminotransferase, namely GGT (EC 2.6.1.4). The Asp-family pathways are critically important for plant nutrition and stress regulation due to synthesizing four essential amino acids, including Lys, Thr, Met, and Ile, initially involving mono-functional Asp kinase (AK, EC 2.7.2.4) and dual-functional Asp kinase-homo-Ser dehydrogenase (AK-HSDH, EC 1.1.1.3). These two enzymes derive the formation of aspartyl-4-phosphate (Asp-4-P), which is further metabolized into quinolinic acid as a central precursor [16]. Asp drives arginine (Arg) biosynthesis controlled by a small family enzyme, so-called argininosuccinate synthase (ASS, EC 6.3.4.5), which combines ligation with citrulline into argininosuccinate, a final substrate for Arg biosynthesis [17]. NAD is a ubiquitous coenzyme involved in oxidation-reduction reactions with functions in all organismal metabolic reactions and cell regulation. Serving as the essential molecule for the de novo biosynthesis route to NAD, Asp is irreversibly converted into α-iminosuccinate by Asp oxidase (AO, EC 1.4.3.16) at an early step in many bacteria and plants [18]. Pyrimidine de novo biosynthesis starts from Asp transcarbamoylase (ATC, EC 2.1.3.2), catalyzing the condensation of carbamoyl Asp from Asp and carbamoyl phosphate, and the latter is released from the hydrolyzed Gln [7].

Forest trees exert significant roles in the overall homeostasis of ecosystems and have economic importance in wood-derived materials, including biofuels, paper pulp, and various metabolites. Sustainable development of forest resources is needed to satisfy the increasing demand for forest-derived products and to preserve natural forest stands [19]. The fine-tuned Asp metabolic pathways are associated with accumulation of N reserves during dormancy, N remobilization, uncoupled N uptake by roots in seasonal growth and recovery from disturbances, such as browsing damage [20]. The current knowledge of molecular mechanisms of Asp metabolism underlying forest growth, development, and ecological adaptation remains a limitation in trees. The expanding repertoires of multiple omics, and advancements in metabolic engineering, provide novel insight into addressing the critical biosynthetic and regulatory processes regarding health threat in tree physiology [21,22]. Extensive efforts have attempted to fortify the biosynthesis of Asp and its transformed amino acids, which play a central function in N metabolism and influence multiple cellular pathways, both in primary and secondary metabolism [23].

The multiple actions of Asp being a critical hub to drive diverse metabolic pathways and the bioactive molecules to mitigate stress cues have been reviewed recently, prompting the requirement for urgent work to profile enzymes in Asp metabolic pathways and to understand how Asp networks influence opposing virulence and defense mechanisms in forest trees [24]. Hitherto, rare research profiled the molecular patterns of isoenzymes in Asp metabolic pathways and associated metabolites required for defense responses. Also, the intimate relationship between responsive isoenzymes in Asp pathways and altered Asp-derived metabolites remains poorly unknown in *Populus* under exposure of *Fusarium solani* (Fs), a root-infecting fungal pathogen. The research objective is to use the transcriptomic atlas, combined with metabolomics assays, to determine the molecular profiling of encoded isoenzymes and differential metabolites flux within Asp metabolic pathways. These findings disclosed the potential of Asp metabolic pathways involving defense response and immunity in *Populus*. Moreover, our work provides new insights for unraveling of the physiological roles of specific isoenzyme(s) and associated metabolites in the reinforcement of defense mechanisms in trees.

## 2. Results

### 2.1. The Populus Candidate Isoenzymes Involved in Asp Metabolic Pathway

Asp centralized metabolic pathways to different branches of critical metabolites involve at least 11 prominent enzyme families (Appendix A). To identify the candidate genes in *Populus*, systematic pBLAST searches, using homologous sequences from *Arabidopsis* as queries, were conducted against the *Populus* genome (v3.1). After removing redundant sequences and manual reannotation, forty-four putative isoenzymes were designated following the nomenclature proposed on the chromosome (Chr). Basic molecular information of the gene ID, genomic sizes and Chromosomal locations, coding sequence (CDS), and open reading frame (ORF) were presented, along with the theoretical molecular weight (MW) of amino acids (aa), isoelectronic points (pI), and in silico prediction of the subcellular target (Appendix A). These enzyme isogenes in Asp metabolic pathways are composed of eight multi-gene families, including nine GSs, four GOGATs, ten AspATs, three ASs, five Aspgs, four AlaATs, two AKs, four AK/AK-HSDHs, two ASSs, and three single-gene families (AsnAT, AO, and ATC). Along with 31 homologs in *Arabidopsis*, multiple full-length sequences alignment was processed to assess the evolutionary relationship. As shown in Figure 1a, forty-four *Populus* isoenzymes were categorized into two major groups, subgroup I and subgroup II, including AspAT, AK/AK-HSDH, ASS, and ATC families. Subgroup I comprised GS, GOGAT, AS, PAT, AlaAT, Aspg, and AsnAT families, appearing to be divided into two clades. Further collinearity analyses revealed that 30 homologous gene pairs existed between *Populus* and *Arabidopsis* genomes (Figure 1b).

The retrieved forty-four *Populus* isogenes showed an uneven distribution among 19 of 20 chromosomes except for Chr 11 (Figure 2), based on mapped chromosomal locations. The gene clusters identified with more than two single members were predicted on Chr 1, 5, 6, 8, 10, and 14. The deduced genomic structures revealed high conserved exon/intron organization patterns within each enzyme family (Appendix A). The upstream promoter sequences of the isogenes were input in PlantCARE to predict the stress-related *cis*-elements, including anaerobic induction (ARE), drought-responsiveness (MBS), defensiveness, stress-responsiveness (TC-rich repeats), wound-responsiveness (WUN-motif), and low-temperature responsiveness (LTR). Besides this, the response elements involved in phytohormones, including MeJA (CGTCA/TGACG), ABA (ABRE), ethylene (ERE), GA (GARE/TATC-box), and auxin (TGA/AuxRR-core), were analyzed (Appendix A). The documented elements on specific gene promoters contributed to identifying the environmental and phytohormone cues that modulate spatiotemporal expressions of specific genes.

#### 2.1.1. GS, AsnAT, AO, and PAT

The *Populus* genome database demonstrated seven cytosolic GS and two plastidic GS genes (*PtGS4* and *PtGS5*). Phylogenetic analysis between *Populus* and *Arabidopsis* also clustered fourteen GS encoding genes into two groups composed of the cytosolic and plastic types (Figure 1a). These GS isoforms were mapped to separate chromosomes, except for *GS8* and *GS9*, with the exact location on Chr 17. Six of seven cytosolic GS genes showed an identical length of ORF and aa but differed in genomic size and organization, comprising 10-12 introns. Two plastic GS genes were comprised of 12 introns, encoding predicted mature GS proteins of 47.75 kDa (PtGS5) and 47.89 kDa (PtGS6). All cytosolic GS proteins had varied protein MW, from 18.04 kDa (PtGS8) to 39.26 kDa (PtGS1). The depicted genomic structures revealed that the GS family is wholly duplicated in the *Populus* genome, which may involve the functional homeostasis of N metabolism related to Glu-derived metabolite biosynthesis [25].

In the *Populus* genome, a single gene was designated as AsnAT1 protein, and it was mapped to the location on Chr 1 (Figure 2). The genomic structure revealed that *PtAsnAT1* contains five introns encoding a protein predicted MW of 44.08 kDa (Appendix A). PtAsnAT1 shows a high protein sequence identity to *Arabidopsis*, displaying a strictly conserved pattern with other crops and moss [26]. In addition, AO was encoded by a single gene designated as *PtAO1* and located on Chr 1. The phylogenetic tree revealed that homologous AO clustered with AsnAT in one subclade. Like the *Arabidopsis AtAO* (*FIN4*), PtAO1 predicted a protein MW of 71.70 kDa, targeting the plastid. The PAT protein sequence and biochemical properties were more divergent to other eukaryotic AspATs from plants and animals than cyanobacterial [12]. Two *Populus* prokaryote-type PATs were identified with *PtPAT1* and *PtPAT2* on Chr 5 and Chr 7, respectively (Appendix A). The genomic organization revealed that *PtPAT* contained nine introns encoding plastidic proteins with predicted MW of 51.59 kDa (PtPAT1) and 51.89 kDa (PtPAT2), which were larger than other eukaryotic AspATs.

#### 2.1.2. GS, AsnAT, AO, and PAT

The *Populus* genome database identified four putative GOGAT isoenzymes, including two Fd-GOGAT encoding genes, *PtGOGAT1* and *4*, and two NADH-GOGAT encoding genes, *PtGOGAT2* and *3* (Appendix A). The phylogenetic tree revealed that the homologs between *Arabidopsis* and *Populus* were categorized into two clades within subgroup I (Figure 1a). Four GOGAT-encoding genes were mapped to locations on separate chromosomes. Interestingly, two pairs of identified NADH-GOGAT and GS encoding genes were clustered on Chr 12 (*PtGOGAT2* and *GS6)* and Chr 15 (*PtGOGAT3* and *GS7*). For Fd-GOGAT, *PtGOGAT1* and *4* displayed similar genomic structures, composed of 32 introns with predicted protein MW of 162.7 kDa and 176.5 kDa, respectively. By comparison, both homologous NADH-GOGAT isogenes (*PtGOGAT2* and *3*) had the same ORF and aa sequences length, containing 21 introns, but a higher predicted protein MW of 243.90 kDa for PtGOGAT2 and 244.10 kDa for PtGOGAT3. The Fd-GOGAT and NADH-GOGAT isoforms shared an approximate 80% identity in the protein sequence.

AlaAT isoforms in the *Populus* genome with the deduced compartmentation in the peroxisome (PtAlaAT1, 3, and 4) and mitochondria (PtAlaAT2) are in line with a previous report [27]. These gene isoforms were mapped to separate chromosomes. Two AlaAT isogenes were localized on Chr 1 (*PtAlaAT1*) and Chr 3 (*PtAlaAT2*), and the rest on Chr 8 (*PtAlaAT3*) and Chr 10 (*PtAlaAT4*). The analyses of conserved genomic patterns revealed *PtAlaAT1* and *2* were composed of 14 introns, while 12 introns were identified for *PtAlaAT3* and *4*. Three peroxisomal AlaAT showed an identical sequence length of ORF and aa but varied slightly in the predicted MW from 52.98 to 53.48 kDa, while the mitochondrial PtAlaAT3 showed a higher MW of approximate 58.27 kDa. The phylogenetic tree revealed a close relation to the clustered AS family in one subclade (Figure 1a).

The homologous search in the *Populus* genome led to three AS encoding genes identified with subcellular localization to cytosols. *PtAS1* comprising eight introns for a predicted protein MW of 65.5 kDa was mapped on a location in the middle region of Chr 1. *PtAS2* and *3* comprised 13 introns (Chr 5) and 12 introns (Chr 9) and encoded predicted protein MW of 65.64 kDa and 66.07 kDa, respectively (Appendix A). However, PtAS1 and PtAS3 shared a higher overall protein sequence identity (>92%) than PtAS2, which clustered AtASN2 and AtASN3 together in the phylogenetic tree.

Five *Populus* Aspg encoding genes were identified to show various deduced subcellular targets (Appendix A). The phylogenetic tree revealed that the homologous Aspg were clustered into one subclade within the subgroup I, except for PtAspg3, which appeared close to the AlaAT family (Figure 1a). Only *PtAspg1* was mapped on Chr 2, and four other isoenzyme genes were located on Chr 14. *PtAspg1* and *2* showed a conserved genomic structure. They contained four introns, consisting of 3038 bp and 1967 bp nucleotides, and coded a similar predicted protein MW of 34.68 kDa and 34.60 kDa, respectively. Likely, *PtAspg3* and *4* comprised four introns and consisted of 2919 bp and 2705 bp nucleotides, coding a predicted protein MW of 37.90 kDa and 37.93 kDa, respectively. The biochemical study revealed that mature Aspg enzymes were heterotetramers, comprising two different subunits, processed by a conserved autoproteolytic cleavage site of precursor, exposing an N-terminal catalytic nucleophile [28].

#### 2.1.3. AspAT, AK/AK-HSDH, ASS, and ATC

The mature AspAT enzyme forms a homodimer in higher plants and is present as a family of isoforms, targeting different subcellular compartments (Appendix A). Analysis of chromosomal distribution featured four AspAT isoforms with their predicted locations on Chr 6. The *PtAspAT3* and *4* showed localization to cytosols, and *PtAspAT2* and *5* were localized to mitochondria and plastids, respectively. The plastic *PtAspAT9* and cytosolic *PtAspAT10* were both mapped to Chr 18. The *AspAT7* was predicted a location on Chr 14, thought to be localized to mitochondria. Based on our previous report, phylogenetic analyses of full-length AspAT homologs between 12 species revealed the presence of two significant subfamilies, namely eukaryotic Iα (A, B, and C) and prokaryotic Iβ. The latter displayed a close relation to the deduced subcellular compartmentation [29].

In higher plants, a small family gene encodes AK and AK-HSDH enzymes. The *Populus* genome contains two AK encoding genes (*PtAK1* and *2*) and two AK-HSDH-encoding genes (*PtAK-HSDH1* and *2*). Both *PtAK1* and *2* were mapped to separate chromosomes with conserved genomic structures (12 introns) and predicted protein MW of 61.05 kDa and 61.71 kDa, respectively. By contrast, two AK-HSDH encoding genes had 17 introns, showing a higher protein MW of 100.80 kDa for PtAK-HSDH1 and 100.33 kDa for PtAK-HSDH2. The phylogenetic analysis revealed a close relationship between the homologs and clustered AspAT family genes in one subclade (Figure 1a). When searching for the *Populus* genome, *PtASS1* and *2* showed a high protein sequence identity, targeting plastids (Appendix A). They showed conserved genomic patterns, containing nine introns with the predicted protein MW of 54.25 kDa and 54.35 kDa, respectively. The plastidic ATC was encoded by a single gene designated as *PtATC*, comprising five introns with a predicted protein MW of 42.94 kDa (Appendix A). The phylogenetic tree revealed a divergent relation for the homologous ATC.

### 2.2. Identification of Enzyme Isogenes with Root-Specific Expression Patterns

Forty-four identified isoenzymes in the Asp metabolic pathway were examined in the source and sink tissues, using transcriptomic sequencing (RNA-seq) to explore gene expression patterns, particularly in the roots. RNA-seq data were available from the Phytozome (v13.1) database, demonstrating a significant variation of transcript abundance in seven selected vegetative tissues, including the root tip (RTP), root (RT), stem internode (SI), stem node (SN), leaf young (LY), leaf immature (LM), and leaf expanded fully (LEF). Based on Figure 3a, approximately 20 gene isoforms displayed the transcript abundance in either roots or the root tips, while more than 17 gene isoforms showed high expression levels in the leaves. However, at least 15 genes appeared to have no detected transcript abundance in all vegetative tissues. Significant root-specific expression patterns were identified for *PtGS3*/*4*, *PtAS1*, *PtAlaAT1*, and *2*, while some genes (*PtGS7*, *PtGOGAT1*/*3*, *PtPAT1*, *PtAspAT10*, *PtAspg2*, and *PtASS1*) showed distributed expression in all selected tissues, but it was more dominant in the roots. It was found that *PtGS2*, *6*, and *9* were explicitly expressed in roots and stems. At least six isoenzyme genes appeared to be explicitly expressed in root tips and mature leaves, including *PtGS4*/*5*, *PtGOGAT1*, *PtAspAT9*, *PtAlAT3*, and *PtAS3*. *PtAspg1* displayed the specific expression in the stem. Interestingly, four isoenzyme genes (*PtGS1*, *PtAS3*, *PtAlaAT4*, and *PtATC*) showed an even transcript distribution in stems and leaves, whereas this was insignificant high in the roots.

Forty isoenzyme genes that identified genes transcript abundance in RNA-seq data were further validated by quantitative real-time PCR (qRT-PCR), using the selected tissues (young leaves, YL, mature leaves, ML, stem, ST, and roots, RT) of in vitro cultured plants (Figure 3b). The experimental validation confirmed that most GS family genes revealed enriched expressions in the roots, except for *PtGS1*, *3*, and *8*. As examined by RNA-seq, two NADH-GOGATs, *PtGOGAT2* and *3* displayed a specific expression in the roots, while two Fd-GOGAT gene isoforms were significantly expressed in the leaves. Likewise, in the AlaAT family, *PtAlaAT1* and *2* showed root-specific expression patterns, while *PtAlaAT3* and *4* were highly expressed in the leaves. The AspAT families were universally expressed in each selected tissue, and five members (*PtAspAT1*, *2*, *4*, *5*, and *10*) showed significant expression levels in the roots combined with the cycle of threshold (CT) values. For the AS family, expression of all three enzyme isoforms was detected in the roots, particularly for *PtAS1* and *2*. Only one member (*PtAspg2*) exhibited more significant expression levels in the roots, even if the transcripts were detectable for other Aspg isoforms. *PtAK1* and *PtAK-HSDH2* appeared to be more abundant in all tissues than other isoforms within the AK/AK-HSDH family. *PtASS1* and *2* displayed an even expression distribution in the roots and leaves, in line with the RNA-seq data. Besides these findings, within the single enzyme families, *PtATC* was inspected to show a higher expression level in the roots than *PtAsnAT1* and *PtAO1*, with a few discrepancies observed in RNA-seq. Nevertheless, the gene expression levels, validated by qRT-PCR, depicted almost consistent patterns in the transcriptomic data, prompting approximately 26 isogenes to be revealed with abundant expression in the roots.

### 2.3. Transcriptomic Survey of the Fs Responsive Isogenes in Populus Roots

A previous report has revealed a whole RNA-seq transcriptome used to explore incompatible interactions in *Populus* roots with a time-course infection of pathogenic Fs [30]. The differentially expressed genes (DEGs) were documented during hours of post-inoculation (hpi) with Fs. Based on the transcriptome atlas, the enzyme isoforms with examined transcript abundance (FPKM, fragments per kilobase per million mapped fragments) in Fs infected roots were comparatively surveyed between 0, 24, 48, and 72 hpi (Figure 4a). At 24 hpi, only five DEGs (e.g., *PtGS1*, *PtGOGAT4*, *PtAspAT5*, *PtAS1*, and *PtAS3*) showed the transcript induction, and two DEGs (*PtAlaAT4* and *PtASS2*) appeared to be suppressed compared to 0 hpi (control). While the number of the highly DEGs increased to 15 after 48 hpi, in contrast to the control, suggesting successful colonization of Fs to the host roots. At 72 hpi, the total number of DEG decreased to 10, which may be due to the initiation of visible root necrosis caused by Fs toxic infection [30]. Among the DEGs (FPKM > 10) depicted during plant-pathogen interactions, approximately 14 critical DEGs in the Asp metabolic pathway revealed consistently altered values.

Given findings of identified DEGs with significantly changed values in RNA-seq, the qRT-PCR was used to further evaluate the gene expression patterns during the four-time courses of Fs infection. The validated results by qRT-PCR were compatible primarily with the transcriptomic sequencing data, confirming 13 DEGs with significantly induced expression in response to Fs infection. These Fs-responsive isoenzymes in *Populus* roots were composed of eight families, including 12 genes (*PtGS2/6*, *PtGOGAT2*/3, *PtAspAT2/5/10*, *PtAS2*, *PtAspg2*, *PtAlaAT1*, and *PtAK1*) showing up-regulation and one down-regulation for *PtAlaAT4* (Figure 4b). These isoenzyme genes verified that an increase in expressions during Fs infection courses might result from the stress- or signaling-related regulatory elements (e.g., wounding, defense and stress response, and phytohormone) at the gene upstream (Appendix A).

### 2.4. Perturbation of the Amino Acids and Derivatives in Fs Infected Roots

The significantly altered expression of isogenes involving Asp metabolic pathways allowed us to examine the related metabolites profile upon Fs infection. The contents of differential free amino acids and derivatives in Fs infected roots were quantified using the HPLC assay and non-targeted metabolomics. The HPLC quantification demonstrated that Glu, Arg, Asp, Asn, and Ala appeared to be the predominant proportions within the pool examined throughout the entire time-course of Fs infection (Figure 5a). The variable metabolites were not significantly detected at 24 hpi compared to the control (0 hpi), except for Ala, Thr, and phosphatidylethanolamine, showing a statistical increase or decrease in levels. When the Fs infection time was prolonged to 48 h, a more enriched accumulation of amino acids and derivatives was inspected at 48 hpi. Notably, the 48 h of Fs infection led to increments of 2.52-fold of Glu, 2.60-fold of Asp, 2.30-fold of Asn, 2.56-fold of Ala, and 2.05-fold of proline, 3.38-fold of carbamide. The differences in levels between the histidine, Ser, valine, and α-/γ-aminobutyric acid (A-/GABA) appeared insignificant. In contrast to 48 hpi, the prolonged Fs infection time resulted in markedly elevated Glu, Asp, proline, and A-/GABA levels. A slight statistical drop was observed for Asn and Ala at 72 hpi (Figure 5a).

Furthermore, the altered accumulations of amino acid-related metabolites in Fs infected roots were analyzed by a non-target metabolomics assay. Since very few significant changes were detected during 24 hpi in roots using HPLC, the non-targeted metabolomics analyses focused primarily on identifying the differential metabolites between the compared groups (48/0 hpi, 72/0 hpi, and 72/48 hpi). A total of 3029/304 metabolites were identified with annotation using the liquid and gas chromatography-mass spectrometry (LC-/GC-MS) dual platforms, of which 162/42 were categorized into amino acids and derivatives subclass. When the criteria, variable importance of projection (VIP) >1 and *p* < 0.05 were set up, the numbers of various metabolites in different comparisons were determined, showing 26 in 48/0 hpi, 24 in 72/0 hpi, and 9 in 72/48 hpi (Appendix A). Based on the relative abundant values, eleven amino acids and derivatives were detected with significant changes at three-time courses of Fs infection (Figure 5b and Appendix A). The non-targeted metabolomics analyses, combined with the HPLC assay, confirmed Glu, Asp, Asn, proline (Pro), and GABA responding to Fs infection.

## 3. Discussion

### 3.1. The Functional Roles of Isoenzymes in the Asp Metabolic Pathways

The *Arabidopsis* cytosolic GS genes are expressed in the vascular tissues, suggesting a fundamental role in N remobilization of senescent leaves and primary NH_4_^+^ assimilation in roots [31]. The cytosolic *GLN1;2* gene knockout mutants exhibited impaired seed developments, while new findings indicated involvement in NH_4_^+^ detoxification and N assimilation under a high nitrate supply, confirmed in maize [32]. In rice, a cytosolic *OsGS1;1* modulated central metabolite homeostasis and plastid development in roots via reverse genetics combined with multi-omics profiling [33]. The loss-of-function mutants of cytosolic *OsGS1;2* caused a severe impact on lignin deposition and tiller development cessation, suggesting a role in maintaining the balance of amino acid and sucrose metabolism [34]. GS transcripts and activities showed significant promotion upon various stress cues, reflecting different functional patterns [35]. Constant induction of *OsGS1;1* and plastic *OsGS2* in rice improved agronomic performance, including tolerance to abiotic stresses (e.g., osmosis and salinity) and resistance to herbicides [36]. Consistently expressed cytosolic GS (*HvGS1-1*) enhanced N utilization efficiency and grain yield and protein quality under elevated CO_2_ in barley and other crops [37]. The transgenic hybrid *Populus* increased tolerance to drought stress and herbicides by ectopic overexpression of cytosolic GS genes [38,39]. The significantly improved N use efficiency and biomass observed in the above mutants suggested essential roles of GS in N translocation from root to shoot and the biotechnological relevance to phytoremediation of N pollution [40,41,42]. However, much remains unknown about the physiological role of GS in defense modulation. The highly responsive patterns of *PtGS2* identified upon Fs infection may contribute to the defense regulation of N metabolism homeostasis associated with varying biotic stress factors, including the fungi pathogen.

Our qRT-PCR evaluation demonstrated that *PtGOGAT1* was the major Fd-GOGAT gene abundantly expressed in source leaves. In comparison, two NADH-GOGAT genes, *PtGOGAT2* and *3*, showed transcripts at high constitutive levels in non-photosynthetic roots, in which their significant promotions were identified upon Fs infection (Figure 4b and Figure 5b). It has been postulated that the cytosolic GS and NADH-GOGAT cycle is involved in NH_4_^+^ assimilation primarily in roots where the fixed N is translocated from roots to leaves through the vascular bundle [43]. Notably, two pairs of clusters (*PtGS6/PtPtGOGAT2* and *PtGS7/PtPtGOGAT3*) displayed chromosomal co-localization and similar tissue-specific expression patterns, suggesting the GS/NADH-GOGAT cycle in regulating NH_4_^+^ assimilation (Figure 2 and Figure 3b). Previous research revealed that *Arabidopsis* lacking *Fd-GOGAT* expression led to chlorotic phenotypes, concurrent with altered amino acids and chlorophyll biosynthesis [44]. The *NADH-GOGAT* deficient mutants displayed reduced biomass and chlorophyll and Glu contents compared to the wild type [45]. These findings imply that both Fd- and NADH-GOGAT activities are independently required for NH_4_^+^ assimilation and Glu metabolism, having a tight association to seed filling and yield control [46,47]. GOGAT is the critical enzyme involved in the *de novo* biosynthesis of acidic Glu, which orchestrates crucial metabolic, homeostatic, and signaling functions with pivotal roles in plant stress and defense adaptation [48,49]. The *Arabidopsis* Fd-GOGAT isoform (*GLU1*) modulated iron homeostasis and cadmium tolerance by affecting Glu biosynthesis. The plant mutants displayed decreased iron levels in the shoots and severe leaf chlorosis under the Fe starvation conditions [50]. In rice, a mutant in the *Fd-GOGAT1* (*Lc7*) showed insensitive behavior in leaves to seven *Xanthomonas oryzae* pv. *Oryzae* strains, suggesting the potential actions of Fd-GOGAT in resistance to the bacterial pathogen [51]. However, the underlying defense mechanisms remain unveiled during the plant-pathogen interaction. More recently, a rice blast fungus-derived GOGAT gene (*MoGlt1*) was identified for intermediate native Glu homeostasis critically used for pathogenesis and the regulation of autophagy [52]. This work indicated that the Glu metabolism capacity might be essential for plant host defense and pathogen colonization.

In plants, multiple isoenzymes of AspAT show distinct subcellular targets, including the mitochondrion, plastids, and cytosols. The spatiotemporal expression changes of *AspATs* to varied environmental cues signify functional divergence and physiological roles in regulating C and N homeostasis, facilitating plant growth, development, and stress acclimation [29]. The previous report revealed that *Arabidopsis* mutants, defective in the cytosolic *AspAT2* (*aat2*) or plastidic *AspAT3* (*aat-3*), showed reduced Asp levels and root elongation compared to wild type [53]. By the ectopic expression of a soybean plastic AspAT gene (*GmAspAT5*) in *Arabidopsis*, transgenic lines showed enhanced Asn, Ala, Gly, and Glu contents in seeds where the levels of Asp-family amino acids were decreased, suggesting that excess amino acids are deaminated and metabolized as NH_4_^+^ and C sources [54]. In rice, constantly inducing expression of native plastic or cytosolic *AspATs* led to altered N metabolism and high contents of total free amino acids in seeds, whereas there were no significant effects on plant growth [55]. Silencing the tobacco eukaryotic *NbAsp5* led to a decreased Asn accumulation and much higher levels of lysine. By contrast, suppressing prokaryotic *NbPAT* resulted in severe chlorosis symptoms, impaired growth, a significant reduction in Asn and Phe contents, concomitant with lignin deposition, prompting the central role of plastidic AspAT in N metabolism [56]. Consistent expression of *AtAspAT2* in *Arabidopsis* led to a higher amount of spreading lesions upon necrotrophic fungi pathogen *Botrytis cinerea*, suggesting the involvement of defense regulation through modification of antagonistic substances (e.g., Arg, Pro, and GABA) derived from Glu in favor of nutrient acquisition by the fungi pathogen [57]. Therefore, three AspAT isoforms (*PtAspAT1*, *2*, and *10*) identified strong promotion in response to Fs infection prompt potential roles involved in defense modulation.

The cytosolic ASs/ASNs serve as primary N storage, and transport compounds used to remobilize N between sources and sinks. Previous work in *Arabidopsis* demonstrated that overexpressing lines of the *ASN1* showed promotion in seed protein storage and improved tolerance to N starvation, indicating an efficient translocation of Asn [58,59]. In *Arabidopsis* loss-of-function mutant lines, the disruption of *ASN2* led to a defective vegetative growth concomitant with a suppressed aboveground biomass and low salt tolerance, due to reduction in GABA, Asp, Ala, and Pro accumulation [60,61]. A more recent report revealed that the constant expression of ASN isoform (*OsASN1*) in rice resulted in significantly improved grain yield, protein levels, and N content under N deficient conditions [62,63]. The *OsASN1* knockout mutant lines showed retarded growth phenotypes, tiller number, and bud outgrowth [64]. The entire function of AS in determining free Asn biosynthesis was verified in other crops [65]. These findings underline the physiological importance of ASN/AS in crucial biological processes and in maintaining the homeostasis of NH_4_^+^, recycling, and transport that may influence plant development and stress regulation. Besides, targeting Asn acquisition and utilization has been proved beneficial in various pre-clinical models, shedding light on a broader cancer therapeutic strategy [66]. In *Populus*, three putative AS isoenzymes were functionally characterized using the complementation assay that corroborates our work [67]. Nevertheless, regarding Asn metabolism and stress regulation, the in vivo roles of the AS encoding genes have not been elucidated in *Populus*.

*Arabidopsis* double mutants, deficient in *ASPGA1* and *ASPGB1*, showed an increase of Asn accumulation in mature seeds under high N feeding conditions, but root elongation was halted by Asn, the sole N resource [68]. This work implied that Asn or Asn-derived metabolite biosynthesis was linked to AspAT reaction, rather than to the products of ASPG, Asp, or NH_4_^+^ that initiated root growth inhibition. In soybean embryos, ectopic expression of a common bean *PvAspG2* led to a slight reduction in Asn levels concurrent with a significant increase of Asp contents by 25–60%, but the effects on N levels were variable [69]. Expression of two rice *ASPGs* determined that only *OsASNase2* was abundant in the developing seeds, vascular bundles, mesophyll, and senescent leaves associated with Asp accumulation as inspected amino acids [70]. In comparison with our work, five putative ASPG isozymes were identified in *Populus*, two of which were highly expressed in vegetative tissues, particularly for *PtAspg2*, showing a predominant expression in roots and significant induction responding to Fs infection (Figure 3 and Figure 4). The other three *PtAspg* isoforms may be expressed explicitly in reproductive tissues.

The various expression patterns suggest that AlaAT diversifies biochemical reactions and biological processes throughout the life cycle of plants [71]. Four *AlaATs* exist in *Arabidopsis*, divided into *AtAlaATs* and *AtGGTs*. A novel enzyme gene (At3g08860) was recently endowed with AlaAT activity [72]. *Arabidopsis AlaAT1* knockout mutant accumulated higher levels of Ala upon hypoxia, and less efficiency in utilization of exogenous Ala, indicating the importance of AlaAT in low-oxygen stress recovery and Ala metabolism [73]. Silencing both *AtAlaAT1* and *AtAlaAT2* expressions in RNAi lines showed reduced photorespiratory CO_2_ release and a lower CO_2_ compensation point [74]. The constantly expressing *AtGGT1* in *Arabidopsis* led to a marked increase in free amino acids (e.g., Ser and Gly), and this profile was affected by light and available nutrients [75]. Significant advances in the AlaAT families have primarily focused on N use efficiency in many crop plants. Ectopic introduction of barley *HvAlaAT* in various N-supplied transgenic crops significantly increased the biomass and grain yield, indicating improved N use efficiency [76,77,78]. Root-specific expression of a cucumber *AlaAT* resulted in high N use efficiency in transgenic rice [79]. Unlike herbaceous plants, forest trees have a long lifespan and a perennial woody growth habit with sustained N nutrition by seasonal and internal cycling; therefore, investigating N’s molecular factor and regulatory mechanisms has excellent environmental significance in *Populus*. In our work, four putative AlaAT family members were identified in *Populus*, displaying different spatial-temporal expression patterns in line with a previous report [27].

The mono-functional AK and dual-functional AK-HSDH catalyzed the transfer of Asp to Asp-4-P, then converted it to Asp semialdehyde, the branch-point intermediate used for biosynthesis of limiting Lys or Met, Thr, and Ile, and the committing step leading to the formation of homo-Ser [10]. Here, two AKs and two AK-HSDHs isogenes were identified in the *Populus* genome. Expression quantification revealed that *PtAK1* and *PtHSDH.2* appeared to be more abundant than other isoforms, specifically under the Fs infection conditions (Figure 4). By overexpressing a bacterial feedback-insensitive AK (*bAK*), transgenic tobacco plants showed significantly increased levels of Thr, prompting a functional link between the AK’s catalytic activity and its associated metabolite biosynthesis [80]. Recent research demonstrated that the *Arabidopsis* loss-of-function mutants in the *AK-HSDH2* gene showed increased accumulations of Asp and Asp-derived amino acids, particularly Thr, in four-week-old plant leaves [16]. Increased levels in Asp, Lys, and Met were also observed in either *AKs* and *AK-HSDH1* single knockout mutants, whereas only *AK-HSDH* deficient mutants revealed altered Thr levels, suggesting the importance of AK and AK-HSDH for maintaining the flux and homeostasis of Asp-derived amino acids and feedback inhibition [16,80,81]. In *Arabidopsis*, the *AtAK2* loss-of-inhibition allele mutant perturbed amino acid homeostasis, leading to over-accumulation of Asp-derived amino acids (e.g., Met, Thr, and Ile) and an insensitive susceptibility to the obligate biotrophic oomycete, *Hyaloperonospora arabidopsidis* (Hpa) [82]. This work implied that the metabolic state specifically impedes biotrophic pathogen colonization.

AsnAT/AGT1 was characterized as a peroxisome enzyme, playing a role in transferring Asn into variable amino acids (e.g., Gly, Ala, Ser, and homo-Ser) and 2-OS depending on a wide range of donors: acceptor combinations available for plant metabolism and photorespiration [15]. The previous studies on biochemical and loss-of-function mutants demonstrated that *AGT1* in *Arabidopsis* acted as a Ser: glyoxylate aminotransferase involved in regulating photorespiration and Ser biosynthesis [83]. *Populus AsnAT* showed higher expression levels in leaves than the roots, with no significant response to Fs infection. Previous work demonstrated that the recombinant *Arabidopsis* AGT1 was specifically active to Ser: glyoxylate as a donor: acceptor. The *Arabidopsis* point mutant in *AGT1* led to retarded growth and chlorotic phenotypes compared to wild-type seedlings, which may be due to abolished catalytic activities in vivo [14]. Recently, constant induction of *AtAGT1* in *Arabidopsis* promoted elongation of primary and lateral roots, indicating functional roles of *AtAGT1* involving a complex metabolic network and salt stress tolerance [84].

In plants, the *de novo* synthesis of NAD starts with the oxidation of Asp to α-iminosuccinate catalyzed by the enzyme AO, which is considered pivotal to maintaining NAD homeostasis and signaling molecules in different cellular processes [85]. In *Arabidopsis*, Asp oxidase is encoded by a single gene designated as *AO* or *nadB*, and the recombinant enzyme displays an Asp: fumarate oxidoreductase activity [86]. Likewise, *PtAO1* is the single gene identified in *Populus*, appearing to be insignificantly affected upon Fs infection (Figure 4). NAD represents one of the redox carriers and the cornerstones of cellular oxidation. T-DNA-based disruption of *AO* caused embryo lethal in *Arabidopsis*, suggesting an essential role in plant development [87]. By infecting leaves with a virulent bacterial strain, a significant promotion of *AtAO* transcripts and alternation in crucial Asp-derived amino acids and NAD-derived nicotinic acid provide a clue that higher NAD contents are beneficial for plant immunity by stimulating SA-dependent signaling and pathogen resistance [88]. The AO loss-of-function mutants (fin4) recently demonstrated an impaired stomatal immunity against plant pathogenic bacteria, *Pseudomonas syringae* pv. *tomato*, confirming the *AtAO* specific function in activating NAD-mediated cross-talk with other defense pathways [18].

The plant plastidic ASS converted ligated Asp and non-proteinogenic amino acid citrulline into argininosuccinate, an intermediate substrate linked to the biosynthesis of Arg, then catabolized this to urea and finally degraded it to form CO_2_ and NH_4_^+^ by cytosolic urease [17]. The ASS encoding isogenes remain poorly known in plants. The recent work demonstrated in *Arabidopsis* revealed that a single gene (At4G24830) encoded ASS was characterized in correlation to Arg cycling, but there were no visible effects on growth [89]. A genome-wide association study identified ASS as accounting for 7% of the variation in Arg accumulation in watermelon seeds, implicating functional relevance and potential for marker-assisted selection [90]. Ectopically expressing an *ASS* gene (*argG*) from *Oenococcus oeni* improved acid resistance in *Lactobacillus*, a wine strain for malolactic fermentation [91]. The recent finding suggested that bacterial *ASS* (*argG*) deployed the Arg biosynthetic pathway associated with increased virulence to walnut infection [92]. In mammal cells, the ASS exerts an essential role as the rate-limiting step in Asp metabolism, providing the therapeutical target used for an assortment of metabolic processes and immune responses [93,94]. Nevertheless, the physiological roles of *ASS* have not been well elucidated in plants.

Pyrimidine nucleotides exist abundantly with essential roles in many biochemical and metabolic processes, particularly in nucleic acid biosynthesis and carbohydrate metabolism [7]. The essential enzyme ATC catalyzes the committed step between Asp and carbamoyl phosphate to form N-carbamoyl-Asp involving *de novo* synthesis of uridine-5′-monophosphate (UMP) in plastids (Appendix A). The UMP-ending product of the *de novo* pathway is the precursor for the biosynthesis of all pyrimidine nucleotides [95]. Early biochemical study in wheat-germ revealed that ATCs consisted of a UMP-inhibitory homotrimer without associated subunits, prompting thinking that the catalytic and regulatory sites must reside within the same polypeptide chain [96]. In *Arabidopsis*, crystallization, mutagenesis, and biochemical analyses have recently revealed catalytic and regulatory roles of ATCs involved in modulating *de novo* pyrimidine biosynthesis that is essential for plant growth and development as well as for stress adaptation [7]. In contrast, the physiological significance of ATC is rarely characterized in other plant species, and no available evidence has revealed its role in association with stress adaptation.

### 3.2. Asp and the Asp-Derived Branches of Amino Acids Involved in Plant Defense and Immunity

Host defense responses and induced immunity are mounted with a profound regulation of primary plant metabolism, including biosynthesis of carbohydrates and amino acids and their derived antimicrobial agents and secondary metabolites (e.g., phytoalexin and phytoanticipin) [71]. It is also part of the chemical defense system associated with increased resistance and tolerance to virulent and avirulent pathogens and pathogen-derived elicitors [97]. The altered levels in primary metabolism upon pathogen infection notably influence nutrient resource availability and translocation between sources and sink tissues [98]. Thus, the general recycling of amino acid metabolism derived from enzyme promotion is part of a comprehensive strategy to preserve energy and generate additional energy in catabolic pathways [99]. A survey of coordinated gene networks implicated responsive behavior of primary metabolism in association with energy processing, including glycolysis and the pentose-phosphate pathway, ATP, TCA cycle, and biosynthesis of amino acids required for energy production (e.g., Lys and Met) and photorespiration (e.g., Glu, Arg, Ser, and Gly) responding to various stress cues [100]. Accordingly, as an integral part of the defense and immune systems, amino acid metabolic pathways were presented by observing the modified metabolite composition and release of signaling compounds during pathogen attack [101]. Remarkably, the modifications in steps of Asp and Asp-derived branches of amino acid biosynthetic pathways lead to significant effects on plant-pathogen interactions.

Different N forms regulating physiological response to pathogenic fungal *F. oxysporum* have been demonstrated in cucumber, revealing that NH_4_^+^-supplied plants accumulated more amino acids concurrent with increased disease incidence in roots infected by the pathogen, an indication of the association between amino acid biosynthesis and pathogen colonization [102]. Notably, the amino acid levels in Asp metabolic branches and the expression of catalytic genes in roots showed significant bounce responding to *F. oxysporum* infection [102]. Increasing Asp content in *Arabidopsis* appeared to be correlated with susceptibility to fungi *B. cinerea* [57]. Exogenous homo-Ser increased resistance in a series of *Arabidopsis* mutants (*dmr1*), improved in defense signaling pathways, exhibiting unchanged Asp pathway and downy mildew resistance independent of known immune mechanisms [103].

The Asn metabolism has long fulfilled a critical role in N transport, storage, and stress responses, acting as the primary transport molecule of reduced N in the vascular tissues; also, Asn accumulation showed alternation upon diverse environmental stimuli in different plant species [15]. It has been demonstrated that the metabolism of stress-induced Asn might be associated with adapted stressors, including disease and mineral deficiency, due to active roles in N mobilization and NH_4_^+^ detoxification [104].

Thr-pretreated plants effectively suppressed oomycete pathogen (Hpa) growth *in planta*, leading to a shift back to the AK point mutation that showed resistant phenotypes, proposing variations in Asp pathway regulation [82]. The Lys metabolic pathway generates the immune signal pipecolic acid (Pip), further hydroxylated to N-hydroxypipecolic acid (NHP). NHP acts as an endogenous activator to enhance defense readiness and amplifies response perception, defense priming, and systemic acquired resistance (SAR) for immune mobilization in cases of future pathogen challenge [105]. Based on metabolite analyses in *Arabidopsis*, the levels in Lys concomitant with Pip and α-aminoadipic acid, a Lys catabolite, significantly increased in leaves with SAR-related defense priming to *P. syringae* inoculation. In contrast, the Lys biosynthetic precursor, Asp, appeared to be suppressed [106]. Upon the *P. syringae* infection, the branched-chain amino acids (e.g., Val, Leu, and Ile) were inspected to show a marked accumulation in *Arabidopsis* leaves [107].

Glu pretreatment of pear fruit induced the accumulation of amino acids, particularly in Glu, GABA, and Arg, leading to resistance to *Penicillium expansum* [108]. Aside from N nutrient resources, emerging evidence prompted Glu to be a signaling molecule in defense response and pathogen tolerance through the activation of Glu receptors (GLRs) [49]. Likewise, Glu promoting GLR expression and amino acid biosynthesis resulted in a reduced infection of various pathogens in *Arabidopsis* and crops [48,108,109]. Being compatible with the above findings, in our experiments (Figure 5), the elevated Glu contents in roots support Glu as an immune indicator, mediating the host SAR, defense priming, and local resistance to Fs.

Pro accumulates in many plant species in response to environmental stress (e.g., salt, osmosis, pathogen, and dehydration) and exerts as a signaling molecule in cellular homeostasis, orchestrating specific gene expression used to recover plants from stress stimuli [110] rapidly. Pro can influence stress tolerance in multiple ways, and its metabolism is involved in oxidative burst and hypersensitive response (HR) in correlation with the recognition of avirulent pathogens [101]. Pro-treated cells can diminish reactive and oxidative species (ROS) levels in fungi pathogens, preventing apoptosis [111]. High Pro accumulation in plants can improve the tolerance and antioxidant defense system by maintaining detoxifying enzymes, protein turnover machinery, and activating stress-protective proteins [112,113]. Besides this, previous work revealed that Pro modulated plant defense response to *Agrobacterium*, functioning as an antagonist of GABA-dependent defense, interfered with the GABA-induced degradation of bacterial quorum-sensing signal. This postulation supported the view that plants with a low Pro level displayed reduced tumor symptoms [114]. While, in our experiment, significantly increased levels of Pro observed implied tricky roles in defense response to virulent fungi, possibly resulting in elevated Pro ascribed to the ROS triggered by Fs infection (Figure 5b).

Moreover, exogenous feeding of Arg and Gly increased maize radicle length and lateral root number, suggesting agronomic application in N use efficiency under stress conditions [115]. Asp catabolism leads to Arg biosynthesis, essential for plant defense, supplying precursors for many signaling molecules (e.g., NO and polyamines) against abiotic and biotic stress. The potential benefits of Arg were provided frequently in the biomass and adaptation to multiple stress exposure [116,117,118,119]. However, Arg levels in plant tissues appear to be modulated by complicated mechanisms because most of the experimental manipulations of Arg metabolism did not significantly alter Arg production [17]. Thus, in our case, the physiological response of Arg in defense and immunity seems insignificant, since slightly altered levels of Arg were monitored during the time-course of Fs infection in roots.

## 4. Materials and Methods

### 4.1. Plant Materials, Growth Conditions, and Fungal Infection

Under long-day conditions (25 °C, 16/8 h day/night photoperiod, 50 µE), the *P. trichocarpa* (genotype *Nisqually-1*) was cultured in vitro on a traditional woody plant medium with 30 g L^−1^ sucrose, 0.1 mg L^−1^ IBA, and solidified with 8 g L^−1^ plant agar. According to a previous report, the Fs culture, the spore calculation (1.0 × 10^6^ spores/mL), and the infection on *Populus* roots were conducted [119]. The different tissues and time-course of Fs infected roots (0, 24, 48, and 72 hpi) were collected for transcriptomic sequencing, gene expression, and quantification of amino acids and derivatives.

### 4.2. Sequence Mining, Phylogeny, and Genomic Analyses

Homologs in *Arabidopsis* (TAIR, https://www.arabidopsis.org/, 1 April 2014) were collected as queries to search candidate isogenes in *P. trichocarpa* genome assembly (v3.1) from the JGI gene catalog (https://phytozome-next.jgi.doe.gov/info/Ptrichocarpa_v3_1, v3.1, 30 November 2018) with the E-value cutoff set as 1e-5. The genomic structure was deduced by comparing the CDS and corresponding DNA sequences using the GSDS (http://gsds.gao-lab.org/, v2.0, 31 August 2015) [120]. The chromosomal distribution of gene candidates was obtained from the PopGenIE (http://popgenie.org/chromosome-diagram, v2.2, 6 August 2021). Approximately 1.5-kb upstream regions searched for *cis*-acting regulatory elements in the PlantCARE (http://bioinformatics.psb.ugent.be/webtools/plantcare/html/, 11 September 2000) [121]. The subcellular targeting was deduced by the online programs DeepLoc-1.0 (http://www.cbs.dtu.dk/services/DeepLoc/, v1.0, 7 July 2017) and Phobius (http://phobius.binf.ku.dk/, v1.01, 1 July 2007) [122]. The multiple sequence alignments were analyzed by Clustal (http://www.clustal.org/clustal2/, v2.1, 17 November 2010). The phylogenetic tree was constructed using the MEGA X (https://www.megasoftware.net/, v10.2.2, 1 October 2020) [123]. The evolutionary distances were computed using the Poisson correction method and the number of amino acid substitutions per site.

### 4.3. Transcriptome and Expression Validation by qRT-PCR

Transcriptomic sequencing and data processing were performed on a previous report [30]. The significance of DEGs (FPKM > 5) was judged on the *p* < 0.05, and fold change (FC) of normalized base mean value (*FC* > 2 or *FC* < 0.5) was set as the threshold for significantly differential expression. Both Fisher’s exact test (*p* < 0.05) and multi-test adjustment (false discovery rate (FDR) < 0.05) were applied in DEGs identification. For qRT-PCR validation, RNA extraction and cDNA synthesis were performed according to the previous report [124]. Samples were loaded to a TB green Premix ExTap™Tli RNaseH Plus (Takara, Beijing, China). The mixture was subjected to StepOnePlus™ Real-Time PCR System (AB, Thermo Fisher, Waltham, MA, USA). The primer amplification efficiency was evaluated with dilutions of cDNA, producing an R^2^ ≥ 0.99. The relative gene expression was normalized by the geometric mean of three reference genes (*PtActin*, *PtUBIC*, and *PtEF-1α*) [125]. The primers used for targeting specific genes are listed in Appendix A. The heatmap was constructed using the CIMminer program (http://discover.nci.nih.gov/cimminer/home.do, accessed on 19 July 2018).

### 4.4. Qualitative and Quantitative Analyses of HPLC and Non-Targeted Metabolomics

Four Fs infected roots were harvested and pooled to create one biological sample. For HPLC analyses, free amino acid extraction and detection were performed on a previous report [126]. The lyophilized samples were analyzed by the automatic amino acid analyzer, S-433D (Sykam, Eresing, Germany) in the advanced analysis and testing center (AATC) at Nanjing Forestry University. LC-MS and GC-MS were conducted for non-targeted metabolomics analyses. The accurately weighed samples (80 mg) were transferred to a 1.5 mL Eppendorf tube with two small steel balls added. The L-2-chlorophenyl alanine (0.06 mg/mL, 20 μL) dissolved in methanol was used as the internal standard. The metabolite extraction, analyses, and data processing were performed based on the previous reports [127,128]. For LC-MS, samples were subjected to Dionex U3000 UHPLC with QE plus quadrupole-Orbitrap MS equipped with heated electrospray ionization (ESI) (Thermo Fisher, Waltham, MA, USAA). The metabolic analyses were scanned in ESI’s positive and negative ion mode employed by an ACQUITY-UPLC-HSS T3 column. For GC-MS, samples were analyzed on an Agilent 7890B-5977A MSD system equipped with DB-5MS fused-silica capillary column in a full-scan mode (Agilent, Folsom, CA, USA). The quality controls (QCs) were injected at regular intervals throughout the analytical run to provide a data set to evaluate. VIP values obtained from the orthogonal partial least-squares-discriminant analysis (OPLS-DA) model were used to rank the overall contribution of each variable to group differences. The significance of the metabolites difference (*VIP* > 1.0 and *p* < 0.05) between groups was further verified by a two-tailed Student’s *t*-test.

## 5. Conclusions

Catalytic enzyme-derived metabolic processes modulate plant defense and immune response to pathogens, going far beyond phytohormone cues and cross-talk between the signaling pathways and amino acid metabolism. Asp metabolic pathways generate defense signals, prompting SAR regulators Pip and Lys to catabolite and release natural molecules that exert toxic activity on various microbial pathogens [101]. The Asp-derived cross-talk and conjugation of phytohormones (e.g., JA, SA, and IAA), pyridine nucleotide biosynthesis, Pro metabolism, and oxidation of Arg-derived polyamines in apoplast deploy the yield of profound defense-related compounds involved in metabolic conversion, redox control, phytohormone cross-talk, ROS scavenging, and HR execution [24]. These responsive features modify the host’s defensive capacity, rendering particular immunity resistance to plant pathogens. Recent progress has been made to understand defense responses and immune control regarding the Asp metabolic pathways in various organismal patterns [10,129].

In summary, our work illustrated the critical isoenzymes in Asp metabolic pathways and related metabolite flux as demonstrated by transcriptomic and metabolomics assay. The multi-omics strategy will advance understanding of integrated Asp metabolism, transport, and signal cross-talk in modeling Asp metabolic networks and enzymes targeting a specific tissue or cell type, particularly under pathogenic infection. Besides, the phenotypic screening of the genetically modified lines with over-expression or suppression (or knockout) of specific isogene(s), combined with biochemical analyses and metabolites tracking, have provided alternative ways. The identified isogenes (e.g., *AspAT2*, *5*, and *10*) involved in Asp metabolic pathways with significant responsive features to Fs will be the engineering target for functional analyses in vitro and in vivo under stress exposure. Concurrently, verifying altered accumulation of amino acids (e.g., Glu, Asp, Asn, and GABA) prompted potential to be a defense indicator, aiding in uncovering the synergistically fine-tuned Asp pathway in resistance to fungal disease.

## Figures and Tables

**Figure 1 ijms-23-06368-f001:**
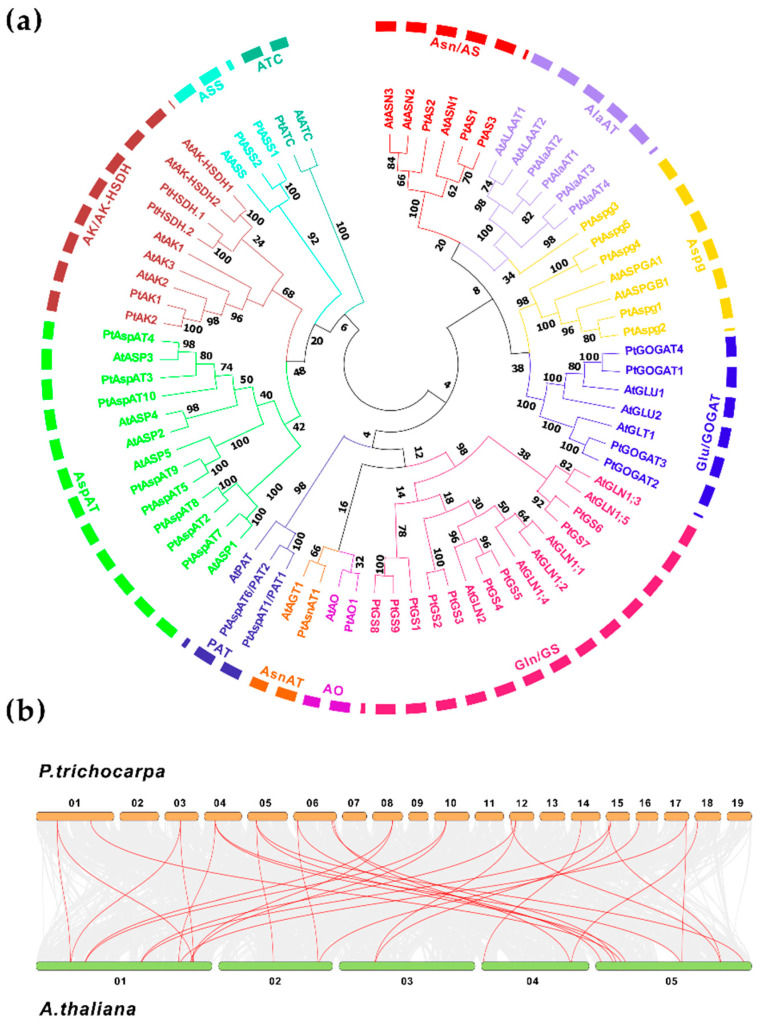
Phylogenetic relationships and collinearity of homologous isogenes between *P. trichocarpa* and *A. thaliana*. (**a**) The phylogeny shows relationships of clustered homologs. (**b**) The analysis shows the collinearity between the homologous isogenes. The unrooted phylogenetic tree was constructed by MEGAX using the Maximum Likelihood methods based on the 1000 bootstrap test.

**Figure 2 ijms-23-06368-f002:**
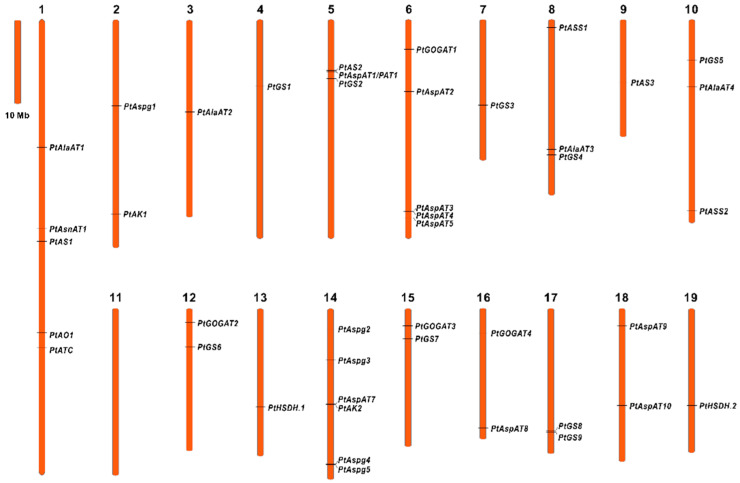
The chromosomal distribution of isogenes involved in Asp metabolic pathways. Forty-four isoenzyme genes were anchored on 19 chromosomes in *Populus* except for Chromosome 11.

**Figure 3 ijms-23-06368-f003:**
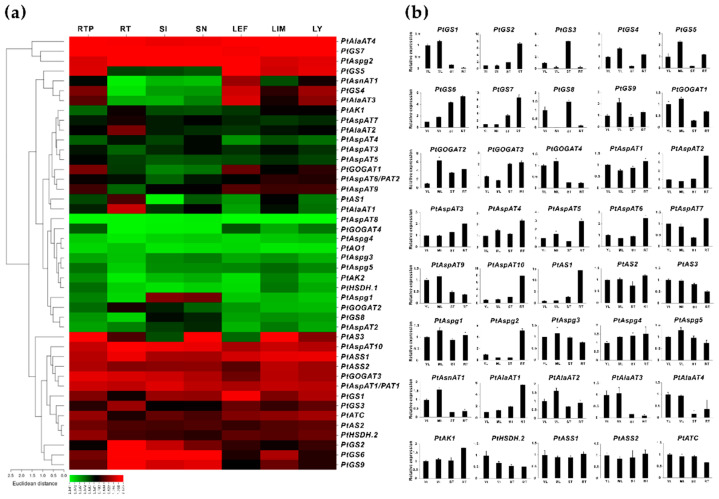
Transcriptomic and expression profiles of isogenes involved in Asp metabolic pathways in various tissues of *P. trichocarpa*. (**a**) The heatmap shows the transcript in vegetative tissues. (**b**) The qRT-PCR evaluation shows the tissue-specific expression. The RNA-seq results were given in the Log of the fragments per kilobase per million reads (FPKM) expression values. At least three independent biological replicates were conducted for qRT-PCR analyses. *PtActin*, *PtUBIC*, and *PtEF-α1* were used as the internal control.

**Figure 4 ijms-23-06368-f004:**
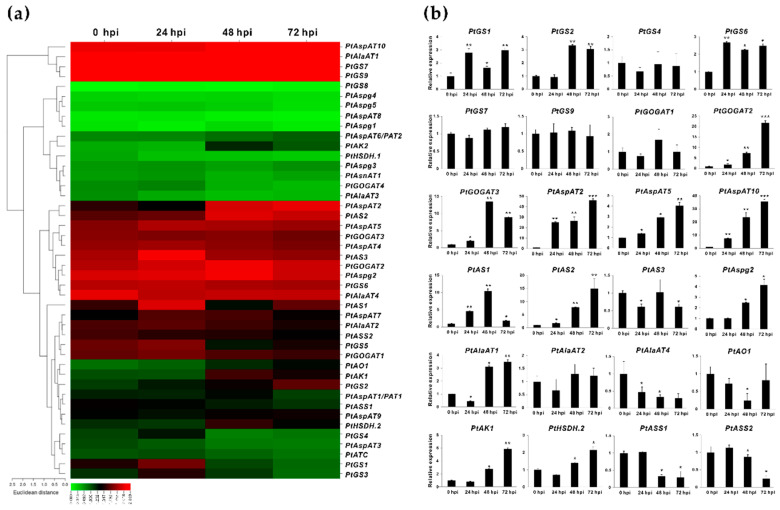
Effects of isogene expression at the time-courses of Fs infection in roots. (**a**) The heatmap shows the transcriptomic profile of isogenes in Fs infected roots (0, 24, 48, and 72 h of post-inoculation, hpi). (**b**) Validation of responsive isogenes compared to the control (0 hpi) by qRT-PCR. The RNA-seq results were given in the Log (FPKM) expression values. *PtActin*, *PtUBIC*, and *PtEF-α1* were set up as the internal control. Data represent mean values standard error (±SE) of at least three independent biological replicates. The asterisks indicate significant differences relative to the control using Student’s *t*-test: *** *p* < 0.001, ** *p* < 0.01, and * *p* < 0.05.

**Figure 5 ijms-23-06368-f005:**
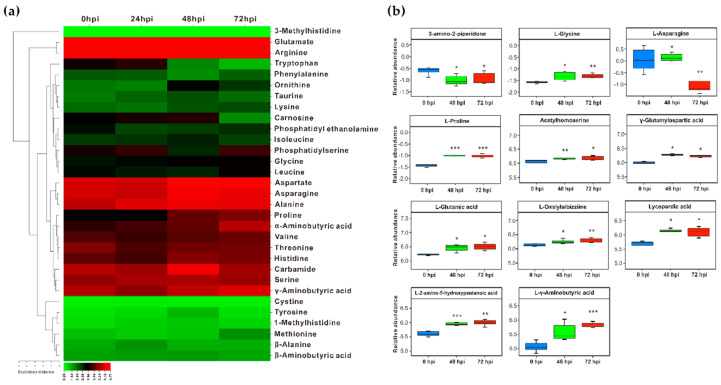
Quantification of the perturbed free amino acids and derivatives in Asp metabolic pathways upon Fs infection in roots. (**a**) The heatmap shows the quantified metabolites by HPLC. (**b**) Relative validation of the significantly altered metabolites, based on the non-targeted metabolomics (LC-/GC-MS) analyses. For the HPLC assay, the Log mean values (n mol/g fresh weight) were used to represent the levels of three independent biological replicates. For non-targeted metabolomics, data represent Log relative mean values based on six independent biological replicates. The asterisks indicate significant differences relative to the control using Student’s *t*-test: *** *p* < 0.001, ** *p* < 0.01, and * *p* < 0.05.

## Data Availability

The raw sequence reads of RNA-seq were deposited in the NCBI with accession BioProject of PRJNA680933 and the accession BioSample, SAMN16927537, including twelve accession numbers of SRR13347970-981 for triplicate data of each Fs treatment (Fs0, Fs24, Fs48, and Fs72).

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
