# Peer review of "New Insight into Aspartate Metabolic Pathways in Populus: Linking the Root Responsive Isoenzymes with Amino Acid Biosynthesis during Incompatible Interactions of Fusarium solani"

_ijms, 2022, doi:10.3390/ijms23126368_

Round 1

Reviewer 1 Report

This in silico study attempted to provide an insight into aspartate metabolic pathways in Populus in response to the incompatible interactions with a fungal pathogen Fusarium solaniThe authors applied multi-omics approaches to examine the responsive isoenzymes and differential amino acids that facilitate aspartate as a cross-talk mediator involved in the metabolite biosynthesis and defense regulation. The findings of this study provide theoretical clues for the in-depth unveiling of the defense mechanisms underlying the synergistic effect of fine-tuned aspartate pathway enzymes and the linked metabolite flux in Populus. However, I have some concerns that need to be addressed before acceptance of this manuscript for publication. 

1) The rationale of the work and objectives are poorly stated in the Introduction section. The figures included in the Introduction section should either move to the results or to the supplementary data.

2) Some of the results may be moved to supplementary data for focusing on the main findings.

3) Discussion section needs to be revised for focusing on the novelty and perpective. 

4) The conclusion section needs to be precisely written with key findings, perspective, and novelty.

Author Response

Reviewer 1:

This in silico study attempted to provide an insight into aspartate metabolic pathways in Populus in response to the incompatible interactions with a fungal pathogen Fusarium solani. The authors applied multi-omics approaches to examine the responsive isoenzymes and differential amino acids that facilitate aspartate as a cross-talk mediator involved in the metabolite biosynthesis and defense regulation. The findings of this study provide theoretical clues for the in-depth unveiling of the defense mechanisms underlying the synergistic effect of fine-tuned aspartate pathway enzymes and the linked metabolite flux in Populus. However, I have some concerns that need to be addressed before acceptance of this manuscript for publication.

1) The rationale of the work and objectives are poorly stated in the Introduction section. The figures included in the Introduction section should either move to the results or to the supplementary data.

R1: Thanks for the good comments! The research objective was strongly addressed in Line 147-154 in the updated ms. Figure 1 has been moved into supplement materials based on the suggestions.

2) Some of the results may be moved to supplementary data for focusing on the main findings.

R2: Excellent comments! After reconsidering the significant results, Figure 3 has been removed from the core structure.

3) Discussion section needs to be revised for focusing on the novelty and perpective.

R3: Thanks! In the present work, the attempt has been made to set up the link between the responsive catalytic enzymes and the respective metabolite altered upon Fs infection. However, owing to the complicated roles and unknown mechanisms of Asp family amino acids (other related amino acids and branch pathways) in signaling control, nutrients regulation, and cross-talk with phytohormone and carbohydrates metabolism, it appeared to be not promising. Nevertheless, the obtained information in our work integrated with more extensive research progress in the discussion may provide a reasonable explanation and postulation within the Asp metabolic network.

4) The conclusion section needs to be precisely written with key findings, perspective, and novelty.

R4: Many thanks! This part has been revised appropriately, and pls check the updated ms. To explore the specific role of the AspAT family in defense and immunity underlying the homeostasis of Asp metabolism is our primary focus. The significantly altered AspAT expressions/activities have been engineered in Populus.

Reviewer 2 Report

line 43: bacteria ?

line 169: ...were...?

line 219: is this reference in the list? (I didn´t find it...)

line 244: ...are...?

line 252: is number of the figure corresponding to the text ??

line 289: maybe to add the explanatory text to the abbreviations of tissues...  

line 358: maybe to add number of the reference.. (29)

line 363 (Fig. 6): ??? I think this figure is identical with fig. 5

line 679: is this reference in the list? (I didn´t find it...)

line 721: italics

Author Response

Reviewer 2:

line 43: bacteria ?

R1: Thanks, this typo has been revised.

line 169: ...were...?

R2: Thanks, this typo has been revised.

line 219: is this reference in the list? (I didn´t find it...)

R3: Thanks, this typo has been revised. Pls check the updated ref 26

line 244: ...are...?

R4: Thanks, this typo has been revised.

line 252: is number of the figure corresponding to the text ??

R5: Thanks, this typo has been revised. Pls, check Line 279 in the updated ms.

line 289: maybe to add the explanatory text to the abbreviations of tissues... 

R6: The detailed description of various tissues has been provided in the text. Pls, check Line 346-347.

line 358: maybe to add number of the reference.. (29)

R7: Thanks, this typo has been revised.

line 363 (Fig. 6): ??? I think this figure is identical with fig. 5

R8: Many thanks, this terrible mistake has been corrected, and the new image (Figure 4) has been inserted into the text.

line 679: is this reference in the list? (I didn´t find it...)

R9: Thanks, this typo has been revised. Pls, check the updated ref 106.

line 721: italics

R10: Thanks, this typo has been revised.

Round 2

Reviewer 1 Report

The authors revised the manuscript satisfactorily.